# Depression and Internet Gaming Disorder among Chinese Adolescents: A Longitudinal Moderated Mediation Model

**DOI:** 10.3390/ijerph20043633

**Published:** 2023-02-18

**Authors:** Mengyun Yin, Shihua Huang, Chengfu Yu

**Affiliations:** 1Department of Applied Psychology, School of Public Health and Management, Guangzhou University of Chinese Medicine, Guangzhou 510006, China; 2School of Psychology, South China Normal University, Guangzhou 510631, China; 3School of Education, Guangzhou University, Guangzhou 510006, China

**Keywords:** depression, internet gaming disorder, maladaptive cognition, mindfulness, adolescents

## Abstract

Internet gaming disorder (IGD) is significantly associated with depression across previous studies, and significantly affects the development of mental health among Chinese adolescents. In this two-wave longitudinal research, we tested the mediating role of maladaptive cognition and the moderating role of mindfulness in the linkage between depression and IGD among Chinese adolescents (*N* = 580, 355 females, average age = 15.76 years, *SD =* 1.31) who completed questionnaires. Results of regression-based analyses showed that depression was positively related to IGD. Maladaptive cognition significantly mediated the link between depression and IGD. Moreover, mindfulness moderated the second part of the mediation process. Specifically, as the level of mindfulness increased, the influence of depression on the future IGD through maladaptive cognition was weakened. The present study demonstrates the key roles of maladaptive cognition and mindfulness in the link between depression and IGD, and further supports the cognitive–behavioral model of pathological Internet use.

## 1. Introduction

Video games are popular in our lives; however, excessive use may lead to negative outcomes, especially in the youth population [1]. Adolescents show important developmental changes in physical and mental abilities; however, they have not yet developed full self-control, and it is perhaps for this reason that they are more likely to demonstrate problems associated with Internet gaming disorder (IGD) [2]. Adolescents addicted to online games are prone to problems such as academic regression and negative emotions [3,4,5]. Therefore, it is important to identify factors that influence the development of IGD in adolescents, and the mechanisms by which interventions can reduce IGD in this age group.

To our knowledge, in the exploration of interventive mechanisms for IGD, several studies have been conducted on the crucial relationship between IGD and its related key factors, especially depression and IGD [6], maladaptive cognition and IGD [7], mindfulness and IGD [8], or cognition, mindfulness and IGD [9]. However, few studies have systematically integrated all of these in an independent study to explore the potential connections between them. Additionally, most of the previous studies on IGD were cross-sectional studies, which were not conducive to an in-depth exploration of how the effect of depression on IGD changed over time. Therefore, the present study used longitudinal tracking to test how depression affects IGD over time. Maladaptive cognition was also tested as a mediator, and mindfulness was tested as a moderator in this longitudinal study. The results of the present study may be useful for future interventional treatment of IGD or Internet game overuse (which may not meet the diagnostic criteria for IGD) in adolescents.

In the interventional treatment of IGD, there has been an issue regarding whether IGD can be considered as one of the subcategories of pathological Internet use (PIU) and therefore warrants the use of therapeutic measures. In 2018, the World Health Organization included gaming disorder (GD) as an official diagnostic entity into ICD-11 [10], describing it as a persistent or repeated pattern of playing online or offline games [11,12]. However, due to the rapid development of the Internet, people engage in various online activities (e.g., online game playing, online chatting, and online shopping) using the Internet as a platform; thus, the larger body of research on IGD [12,13,14,15] shows a default view that IGD is classified as one of the subcategories of pathological Internet use, defined as the excessive use of the Internet [12,16]. Additionally, all addictions (including PIU and IGD) share more in common than differences (e.g., tolerance, withdrawal symptoms, conflict, and relapse), which may reflect a common etiology of addictive behaviors [12,17]. Therefore, interventions to treat the symptoms of IGD based on the idea that IGD is a subtype of PIU may be more well-founded and effective.

### 1.1. Relationship between Depression and IGD

Recent studies suggest that IGD may be significantly linked to adolescent depression [18,19]. For example, Cudo et al. [18] found that depressive symptoms were associated with other behavioral addictions such as Facebook addiction and IGD. A longitudinal study conducted by Teng et al. [19] showed that depression positively predicted IGD severity and videogame use six months later. A possible reason for this may be that as online gaming has become the mainstream form of entertainment for adolescents, adolescents who lack self-control tend to play games to alleviate their negative emotions when they are depressed [20]. This pattern in turn increases their risk of becoming addicted to games [18].

From the perspective that IGD is one of the subcategories of PIU, conceptual models of the link between PIU and depression may be helpful for examining the link between IGD and depression. The cognitive–behavioral model of pathological Internet use [21], a reliable model that has been widely used in the research of Internet addiction and IGD, provides a theoretical framework to explain the development and maintenance of problematic Internet use. It holds that depression is a distal predictor of PIU, as individuals with depressive symptoms tend to use the Internet to relieve negative emotions or avoid real-life problems. Consistent with this view, a longitudinal study [22] among adolescents found that depressive symptoms significantly positively predicted the amount of time spent on the Internet one year later. Therefore, based on this model, depressive symptoms may also predict an increase in behaviors associated with IGD, which has been supported by several studies [23,24].

### 1.2. Maladaptive Cognition as a Mediator

Cognition distortion has been shown to play an important role in emotional disorders and addictive behaviors [18,25]. Regarding general Internet use, people who are lonely may become depressed and then develop cognitive distortion, leading to excessive Internet use and eventual PIU [26].

Davis [21] proposed that maladaptive cognition, which can be understood as a person’s incorrect understanding of Internet use and inappropriate expectations of the results of Internet use (e.g., I am worthy of nothing without the Internet [27]), is a proximal and sufficient cause of PIU. This assertion and related empirical evidence suggest that psychopathology (e.g., depression and social anxiety, etc.) indirectly influence PIU through cognitive distortion. Consistent with this view, maladaptive cognitions were found to affect the association between symptoms of emotional disorders and addiction to online games. Individuals with symptoms of depression [28] or anxiety [29] may relieve their negative emotions through gaming online. Playing games online provides instant feedback and rewards, which may bring users a strong sense of control, satisfaction, and worth that may be lacking in real life [30,31]. Game playing may lead to maladaptive cognitions and more engagement in games, eventually increasing the risk of addictive behaviors [32]. Therefore, maladaptive cognition was also proposed to mediate the association between depression and IGD in the present study.

### 1.3. Mindfulness as a Moderator

Based on the cognitive–behavioral model of PIU, Davis [21] emphasized that attention should be paid to maladaptive cognitions in the treatment of PIU. In the path from “depression → maladaptive cognition → IGD”, there may be protective factors that can effectively reduce the risk of IGD [33,34]. Recently, mindfulness-based interventions (e.g., Mindfulness-Oriented Recovery Enhancement [35]) have become popular in the treatment of IGD [36,37]. Mindfulness is the ability to pay attention to one’s experience at the current moment without judgment [38,39]. It is thought to be a protective factor that promotes mental health by improving the individual’s resilience and emotional regulation capabilities [34,40].

Consistent with the ideas espoused by Davis, the results of some previous studies revealed that when mindfulness increases, the biased cognitions are reduced, resulting in less emotional distress such as depression and anxiety [41,42]. In this way, the abilities of self-control and emotional regulation improve [43,44], and the risk of IGD due to negative emotions gradually declines. Therefore, compared with the first part of the “depression → maladaptive cognition → IGD” path, mindfulness as a moderator of the second part of the path may be more important and effective in interventions to prevent and reduce IGD. In other words, mindfulness was hypothesized to significantly moderate the relationship between maladaptive cognitions and IGD in the present study.

Adolescents with high mindfulness may observe and act with awareness, resulting in increased self-regulation and reduction in the compulsion and loss of control related to problematic Internet use [28]. For example, the results of a laboratory study [45] among Chinese college students with problematic Internet use showed that compared to students with low level of mindfulness assessed by a scale, the influence of stress on cue-induced craving for the Internet was buffered among participants with a high level of mindfulness. In an 8-week experiment of Mindfulness-Oriented Recovery Enhancement for IGD, Li et al. [9] discovered that compared to participants in a support group condition, participants who received mindfulness training showed significantly fewer symptoms of IGD, maladaptive cognitions, and craving for online games; these effects lasted more than three months after the experiment. It is possible that mindfulness may reduce maladaptive cognitions related to online games through top-down cognitive control [41], and facilitate positive cognitive reappraisal to become aware of the motivations for gameplay as escapism and venting negative emotions [46], and reform them; thus, as the craving and maladaptive cognitions of playing online games are reduced, eventually the risk of IGD decreases.

### 1.4. The Present Study

This study used the longitudinal tracking method to test the three-month longitudinal connection between depression and IGD in adolescents. A moderated mediation model (see Figure 1) was proposed and three research hypotheses were tested: (1) depression will significantly positively predict IGD; (2) maladaptive cognitions will significantly mediate the association between depression and IGD; (3) mindfulness will significantly moderate the second part of the mediation process (the association between maladaptive cognitions and IGD). Specifically, it will weaken the association between maladaptive cognitions and future IGD.

## 2. Method

### 2.1. Participants and Procedure

We surveyed participants from three secondary schools in the Guangxi province of China at two time points separated by a three-month interval. All participants signed an informed consent form before the experiment. If they wanted to withdraw halfway through, they could leave on their own. Importantly, participants were informed in advance that no data would be disclosed and would only be used for research.

In Wave 1, a total of 591 students between 12 and 19 years between Grades 7 and 11 were invited to participate in the study and completed the questionnaires. In Wave 2, 580 participants (98.14% of the Wave 1 sample) finished the same questionnaires. Data for the present analyses came from the sample that completed both assessments (*N* = 580, 355 females, average age = 15.76 years, *SD =* 1.31). Attrition was mainly because of invalid responses to the questionnaires. Independent sample *t*-tests were used to compare the group with valid data subjects (*N* = 580) to the group with invalid data (*N* = 11) subjects on the study variables. Depression (*t* = 1.89, *p* = 0.09 > 0.05), IGD (*t* = 0.17, *p* = 0.86 > 0.05), maladaptive cognition (*t* = 1.52, *p* = 0.13 > 0.05), and mindfulness (*t* = −0.68, *p* = 0.50 > 0.05) were not significantly different across groups.

### 2.2. Measurements

#### 2.2.1. Depression

Depression was assessed using the Chinese version of the nine-item Patient Health Questionnaire (PHQ-9) [47,48]. Participants reported how frequently during the past two weeks they had the experience described in each item (e.g., “little interest or pleasure in doing things”). They did this using a four-point scale (0 = never, 3 = nearly every day). Responses to each item were summed to a single score ranging from 0 to 27, with higher scores indicating greater depressive symptom severity. Next, the PHQ-9 score was divided into four categories of increasing severity: minimal (0–4), mild (5–9), moderate (10–14), moderately severe (15–19), and severe (20–27). The PHQ-9 has been shown to have good reliability and validity in samples of Chinese adolescents [48]. In this study, Cronbach’s α for the PHQ-9 was 0.92 at Wave 1 and 0.87 at Wave 2.

#### 2.2.2. Internet Gaming Disorder

Internet gaming disorder was assessed with an 11 item scale adapted from Gentile’s Pathological Video Game Use Questionnaire 3, which has been tested with good reliability and validity in a sample of Chinese adolescents [49]. Participants reported whether they experienced the symptoms of IGD over the past six months on a three-point scale (0 = never, 1 = sometimes, 2 = yes). Items included “Do you feel that you pay too much attention to online games?” Responses to each item were recorded as 0 = never, 0.5 = sometimes, 1 = yes, per Gentile’s [3] recommendations. Greater scores indicate greater symptoms of IGD. In this study, Cronbach’s α for the IGD was 0.90 at Wave 1 and 0.92 at Wave 2.

#### 2.2.3. Maladaptive Cognition

The 12-item Chinese Adolescents’ Maladaptive Cognitions Scale (CAMCS) [29] was used to measure maladaptive cognition. The scale mainly involved two parts: thoughts about the self (e.g., “I feel myself more powerful when surfing online”) and the world (e.g., “I don’t have to think about my responsibilities when I am surfing online.”). Participants reported their levels of agreement with these items on a five-point scale (1 = totally disagree, 5 = totally agree). Higher scores indicated more severe maladaptive cognitions related to the Internet. In this study, Cronbach’s α for the CAMCS was 0.93 at Wave 1 and 0.89 at Wave 2.

#### 2.2.4. Mindfulness

Mindfulness was measured with the revised Chinese version of the Child and Adolescent Mindfulness Measure (CAMM) [50,51]. The scale has a total of 10 items with a five-point score of 0–4 (0 = never, 4 = always). All items were reverse-coded, and the higher the total score, the higher the level of mindfulness. In this study, Cronbach’s α for the CAMM was 0.87 at Wave 1 and 0.85 at Wave 2.

### 2.3. Statistical Analysis

To improve statistical power, the missing data were replaced using linear interpolation in SPSS 25.0, then all the continuous variables were standardized. We investigated common method bias (CMB) using Harman’s single-factor test [52,53]. The first factor explained 25.12% of the variance at Wave 1 and 23.43% of the variance at Wave 2, suggesting negligible common method bias [54].

First, we explored descriptive statistics and correlations between study variables. Second, to test our hypotheses, the mediation of maladaptive cognition and the moderating role of mindfulness were tested by using Model 4 and Model 14, respectively, of Hayes PROCESS macro in software SPSS 25.0 [55]. The PROCESS macro has been widely used to test complex models [56,57]. This is because research has found significant differences in IGD between adolescent boys and girls [58,59], and gender is significantly correlated with other variables included in the analysis of the present study (e.g., depression, IGD, mindfulness, etc.). Therefore, gender was a control variable in the path analysis, coded as 0 (female) and 1 (male). An effect is considered significant when the 95% confidence interval (CI) does not include zero, based on a bootstrap random sample (*N* = 5000).

## 3. Results

### 3.1. Preliminary Analyses

Descriptive statistics and correlations among study variables are presented in Table 1. At both Wave 1 and Wave 2, depression was positively correlated with IGD and maladaptive cognition, which coincide with H1, and IGD was positively correlated with maladaptive cognition. However, mindfulness was negatively correlated with depression, IGD, and maladaptive cognition at both Wave 1 and Wave 2.

### 3.2. Testing for the Mediation Model

As shown in Table 2, depression T1 had a significant total effect on IGD T2. When maladaptive cognition T1 was added, depression T1 was positively associated with maladaptive cognition T1 (β = 0.44, *p* < 0.001), which in turn was related to IGD T2 (β = 0.22, *p* < 0.001). Therefore, maladaptive cognition T1 played a partially mediating role between depression T1 and IGD T2 (95% CI = [0.06, 0.13]). The model accounted for 31.75% of the total effect, thereby supporting Hypothesis 2.

### 3.3. Testing for the Moderated Mediation Model

The results (see Table 3) of Model 14 indicated that mindfulness T1 significantly and negatively predicted IGD T2 (β = −0.10, *p* < 0.05). Furthermore, the interaction of maladaptive cognition T1 and mindfulness TI was significantly and negatively correlated with IGD T2 (β = −0.06, *p* < 0.05), suggesting that mindfulness T1 played a moderating role in the b pathway of the mediation model.

To demonstrate how mindfulness T1 moderated the relationship between maladaptive cognition T1 and IGD T2, mindfulness T1 was divided into two groups (high mindfulness and low mindfulness), defined as plus or minus one standard deviation of the average value. Next, we used a simple slopes test to compare the association between maladaptive cognition T1 and IGD T2 at different levels of mindfulness T1. Among participants with low mindfulness at T1 (*M−*1*SD*), maladaptive cognition T1 had a significant, positive effect on IGD T2. Among participants with high mindfulness at T1 (*M*+1*SD*), maladaptive cognition T1 still had a significant, positive effect on IGD T2, but the size of this effect decreased significantly. In other words, as the level of mindfulness T1 increased, the influence of depression T1 on IGD T2 through maladaptive cognition T1 was weakened (see Table 4 and Figure 2), which supports Hypothesis 3. The effect size difference between the high and low groups was significant (95% CI = [−0.10, −0.01]).

## 4. Discussion

The present two-wave longitudinal study constructed a moderated mediation model to explore the underlying mechanisms between depression and IGD. As expected, the results confirmed the mediating effect of maladaptive cognitions in the link between depression and future IGD, and that this effect was moderated by mindfulness. First, consistent with previous longitudinal studies [60,61], we found that depression at its baseline had a direct positive relation to future IGD over the course of three months. This suggests that depression may be a risk factor for the development and maintenance of IGD. Adolescents with negative affect are more inclined to seek recreational activities to rid themselves of restlessness [60]. For example, Teng et al. [19] discovered that adolescents’ depressive symptoms measured by a questionnaire positively predicted video game use and IGD severity in about six months.

Secondly, we found that depression can predict future IGD through the mediating effect of maladaptive cognitions, which indicates that adolescents with depressive symptoms have more maladaptive cognitions about online game use and thus higher risk of IGD. The finding also greatly supports the application of the cognitive–behavioral model of PIU [21] on IGD. Namely, negative affect (e.g., depression) is a distal factor of IGD and may influence IGD by influencing the maladaptive cognitions (proximal factor), such as “I feel the most comfortable online because I don’t have to worry about the exams” and “I don’t have to think about my responsibilities when I am surfing online”. It is possible that adolescents who experience depression may attempt to alleviate their negative emotion by playing online games in which they can quickly and temporarily obtain positive feedback (e.g., escape from negative emotions [62]) and have psychological needs met (e.g., satisfaction with autonomy in daily life and the sense of control [49]). Thus, the inappropriate perceptions of video games and excessive expectation of outcomes after playing games appear and obtain reinforcement by increased engagement in video games; this cognitive distortion will motivate adolescent players to repeatedly seek pleasure and satisfaction by becoming more and more involved in the game, therefore developing and even maintaining the IGD [49,54].

Thirdly, the results showed that mindfulness acted as a buffer against the negative effects of maladaptive cognitions on IGD. Specifically, the relationship between maladaptive cognitions and IGD was found to be weaker for adolescents who had a high level of mindfulness; that is, compared with adolescents with a low level of mindfulness, adolescents with high mindfulness show a lower risk of being addicted to online games, although they might suffer from maladaptive cognitions. According to the Mindfulness-to-Meaning Theory (MMT [63]), mindfulness not only makes individuals aware of their maladaptive cognitions of video games (which may have previously triggered their urge to be overly involved in games), but also enhances control of these cognitive distortions related to Internet games through top-down cognitive control; thereby facilitating cognitive reassessment about motivations for playing games (e.g., escaping from reality and releasing negative emotions [46]) and effectively reducing cravings for games [9,35,45]. In this way, the risk of IGD will be significantly reduced.

The present longitudinal study not only explored the effect of depression on IGD over time, but also systematically integrated the mechanisms of maladaptive cognitions and mindfulness on the relationship between depression and IGD, which complemented previous studies on IGD. Additionally, the results confirm that Davis’ cognitive–behavioral model of PIU is also applicable to IGD, further extending its applicability and validity. Furthermore, the results of this study also suggest a positive effect of interventions for IGD from the perspective that IGD is a subcategory of PIU, with reference to interventions for PIU. This is probably because they are both addictive in nature and have many common features.

However, the present study has some limitations that need to be acknowledged. First, the adolescents who participated in this study were all from Guangxi Province, and game-addicted players may be affected differently in different regions. Therefore, a more diverse set of participants should be studied to extend the conclusions of this research to other groups. Second, maladaptive cognition was tested as a mediator according to Davis’ cognitive–behavioral model of PIU. However, it has been found that there are other psychological factors, such as coping [64] and regulatory focus [65], that mediate the relationship between distal factors and IGD. Therefore, future researchers can study the impact of other psychological factors on IGD, which will also inform the cognitive–behavioral model.

## Figures and Tables

**Figure 1 ijerph-20-03633-f001:**
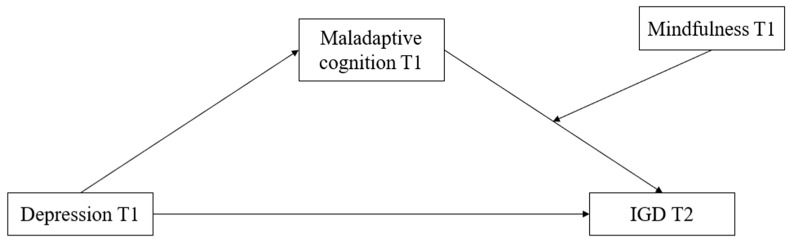
Moderated Mediation Hypothesis Model. *Note.* IGD = Internet Gaming Disorder; T1 = at Wave 1; T2 = at Wave 2. *N* = 580.

**Figure 2 ijerph-20-03633-f002:**
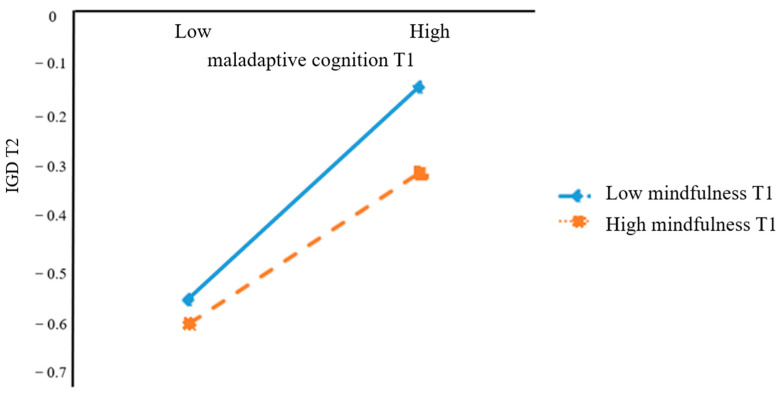
The Moderating Role of Mindfulness in the Relationship Between Maladaptive Cognition and IGD. *Note:* IGD = Internet Gaming Disorder; T1 = at Wave 1; T2 = at Wave 2. Mindfulness T1 was divided into two groups (high mindfulness and low mindfulness), defined as plus or minus one standard deviation of the average value. Furthermore, as the level of mindfulness T1 increased, the influence of maladaptive cognition T1 on IGD T2 was significantly weakened.

**Table 1 ijerph-20-03633-t001:** Descriptive Statistics.

	1	2	3	4	5	6	7	8
1. Depression T1	-							
2. Depression T2	0.51 ***	-						
3. IGD T1	0.36 ***	0.29 ***	-					
4. IGD T2	0.27 ***	0.37 ***	0.63 ***	-				
5. Maladaptive cognition T1	0.43 ***	0.33 ***	0.42 ***	0.34 ***	-			
6. Maladaptive cognition T2	0.31 ***	0.41 ***	0.27 ***	0.45 ***	0.58 ***	-		
7. Mindfulness T1	−0.52 ***	−0.40 ***	−0.34 ***	−0.26 ***	−0.35 ***	−0.29 ***	-	
8. Mindfulness T2	−0.40 ***	−0.55 ***	−0.24 ***	−0.30 ***	−0.31 ***	−0.36 ***	0.53 ***	-
Range	0–27	0–25	0–11	0–9.50	12–60	12–60	0–40	4–40
Mean	5.64	5.63	2.04	2.04	28.85	29.31	21.86	23.26
SD	5.74	4.84	1.97	1.98	9.12	8.61	6.76	6.92

*Note.* IGD = Internet Gaming Disorder; SD = Standard Deviation; T1 = at Wave 1; T2 = at Wave 2. *N* = 580, *** *p* < 0.001.

**Table 2 ijerph-20-03633-t002:** Mediation Analysis.

Independent Variable	IGD T2			Maladaptive Cognition T1			IGD T2		
	β	*SE*	*t*	β	*SE*	*t*	β	*SE*	*t*
Constant	−0.35	0.05	−7.70 ***	−0.09	0.05	−1.86	−0.33	0.04	−7.45 ***
Gender	0.91	0.07	12.35 ***	0.23	0.08	2.99 **	0.86	0.07	11.88 ***
Depression T1	0.30	0.04	8.43 ***	0.44	0.04	11.74 ***	0.21	0.04	5.30 ***
Maladaptive cognition T1							0.22	0.04	5.62 ***
*R²*	0.27			0.20			0.30		
*F*	104.64 ***			71.16 ***			83.98 ***		

*Note.* IGD = Internet Gaming Disorder; SE = Standard Error; T1 = at Wave 1; T2 = at Wave 2. *N* = 580, ** *p <* 0.01, *** *p* < 0.001.

**Table 3 ijerph-20-03633-t003:** Moderated Mediation Analysis.

Variable	Maladaptive Cognition T1			IGD T2		
	β	*SE*	*t*	β	*SE*	*t*
Constant	−0.09	0.05	−1.86	−0.35	0.05	−7.71 ***
Gender	0.23	0.08	2.99 **	0.85	0.07	11.86 ***
Depression T1	0.44	0.04	11.74 ***	0.15	0.04	3.39 ***
Maladaptive cognition T1				0.21	0.04	5.27 ***
Mindfulness T1				−0.10	0.04	−2.55 *
Maladaptive cognition T1 * Mindfulness T1				−0.06	0.03	−2.07 *
*R²*	0.20			0.32		
*F*	71.16 ***			53.00 ***		

*Note.* IGD = Internet Gaming Disorder; SE = Standard Error; T1 = at Wave 1; T2 = at Wave 2. *N* = 580, * *p* < 0.05, ** *p <* 0.01, *** *p* < 0.001.

**Table 4 ijerph-20-03633-t004:** Conditional Indirect Effect at Different Values of Mindfulness.

Moderator: Mindfulness T1	Value	SE	Confidence Interval	
			*LLCI*	*ULCI*
*M−*1*SD*	0.12	0.02	0.07	0.16
*M*	0.09	0.02	0.06	0.13
*M+*1*SD*	0.06	0.02	0.03	0.11
(*M*) *−* (*M−*1*SD*)	−0.03	0.01	−0.05	−0.00
(*M+*1*SD*) *−* (*M−*1*SD*)	−0.05	0.02	−0.10	−0.01
(*M+*1*SD*) *−* (*M*)	−0.03	0.01	−0.05	−0.00

*Note.* T1 = at Wave 1; LLCI = Lower Limit Confidence Interval; ULCI = Upper Limit Confidence Interval; M = Mean; SD = Standard Deviation; SE = Standard Error. *N* = 580.

## Data Availability

All data were available upon request.

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
