# Peer review of "Depression and Internet Gaming Disorder among Chinese Adolescents: A Longitudinal Moderated Mediation Model"

_ijerph, 2023, doi:10.3390/ijerph20043633_

Round 1
Reviewer 1 Report
The paper presents a questioner-based study on 580 Chinese adolescents from the Guangxi province. The adolescents filled questioners at two points in time, separated by three months intervals. Statistical analysis of these questioners, which are the results of the paper, shows that depression was correlated with gaming disorder, and that being more mindful decreases this disorder in the future.
Overall, the paper is well written, very clear in its presentation, writing and flow of ideas. Also, the research method and the analyses that were applied seems good for the purpose of the paper.
My main concerns are with the significance of the research. It seems, as can also be seen from the related work section, that the relationship between depression and gaming disorder is already well known. Also, the ability of mindfulness to decrease depression (and in turn, IGD) is also well established in many papers. So, with that in mind, I significance of the work is not clear in the context of the current literature. In other words, what is the research gap that the authors are trying to fill?
Reviewer 2 Report
Generally speaking this is a clear study which documents the results.
The introduction/review of literature is currently quite sparse and this is where the majority of revision is required to support the development of the paper. Additional framing around PIU as well as IGD is required in this context.
In terms of the methodology, I would ask that the authors document the recruitment process. For example, did students have to self-disclose some level of internet usage? I also feel it would be helpful to document some of the limitations of the study here, before returning to them at the end of the discussion.
